# Neofunctionalization of Glycolytic Enzymes: An Evolutionary Route to Plant Parasitism in the Oomycete *Phytophthora nicotianae*

**DOI:** 10.3390/microorganisms10020281

**Published:** 2022-01-25

**Authors:** Marie-Line Kuhn, Jo-Yanne Le Berre, Naima Kebdani-Minet, Franck Panabières

**Affiliations:** ISA, INRAE, CNRS, Universite Côte d’Azur, 06903 Sophia Antipolis, France; marie-line.kuhn@inrae.fr (M.-L.K.); joelle.le-berre@inrae.fr (J.-Y.L.B.); naima.minet@inrae.fr (N.K.-M.)

**Keywords:** adaptation, evolution, gene duplication, glucokinases, *Phytophthora*, oomycetes, subfunctionalization, virulence

## Abstract

Oomycetes, of the genus *Phytophthora*, comprise of some of the most devastating plant pathogens. Parasitism of *Phytophthora* results from evolution from an autotrophic ancestor and adaptation to a wide range of environments, involving metabolic adaptation. Sequence mining showed that *Phytophthora* spp. display an unusual repertoire of glycolytic enzymes, made of multigene families and enzyme replacements. To investigate the impact of these gene duplications on the biology of *Phytophthora* and, eventually, identify novel functions associated to gene expansion, we focused our study on the first glycolytic step on *P. nicotianae*, a broad host range pathogen. We reveal that this step is committed by a set of three glucokinase types that differ by their structure, enzymatic properties, and evolutionary histories. In addition, they are expressed differentially during the *P. nicotianae* life cycle, including plant infection. Last, we show that there is a strong association between the expression of a glucokinase member in planta and extent of plant infection. Together, these results suggest that metabolic adaptation is a component of the processes underlying evolution of parasitism in *Phytophthora*, which may possibly involve the neofunctionalization of metabolic enzymes.

## 1. Introduction

Oomycetes constitutes a group of lower eukaryotes that comprise of saprobes, as well as highly successful parasites of plant and animals [1]. Historically considered as fungi, on the basis of filamentous morphology, osmotrophic uptake of nutrients, and shared ecological niches, oomycetes are currently placed together with diatoms and brown algae within stramenopiles [2,3]. They are thought to have evolved from marine environments and adapted to a wide range of niches and lifestyles [4]. Oomycetes were proposed to encompass three crown orders (Saprolegniales, Albuginales, and Peronosporales [5]), as well as several orders that are still poorly documented, despite potential ecological impact [4,6,7,8]. Saprolegniales mainly consist of animal parasites, but also few plant pathogens and free-living saprophytes [4]. Albuginales group the obligate, biotrophic parasites of plants that exclusively survive on living tissues. Last, Peronosporales include animal, plant, fungal, and oomycete parasites, as well as a non-pathogenic species. This large order includes several thousands of both formal and provisional species. Among them are some of the most devastating plant pathogens worldwide, including the genera *Phytophthora*, *Pythium* and the intermediate *Phytopythium* genus [9,10], as well as downy mildews, whose taxonomy was recently revised [6,11], and newly discovered organisms that are not known to be pathogenic, such as *Halophytophthora* [12], *Nothophytophthora* [13], and *Calycofera* [7], as well as Salisipiliaceae [8]. Pathogenic species have highly contrasting lifestyles, ranging from a biotrophic, obligate parasitism to necrotrophy, while most *Phytophthora* spp. are hemibiotrophs, in that their infection cycle initiates a biotrophic strategy, followed by a switch to necrotrophy. Collectively, pathogens of the *Phytophthora* genus are responsible for some of the most significant diseases on cultivated plants and natural ecosystems, including the potato late blight (caused by *P. infestans*), sudden oak death (caused by *P. ramorum*), or tobacco black shank (caused by *P. nicotianae*) [14].

*Phytophthora* growth and reproduction involve a complex cycle comprising disseminating, reproductive, resting, and invading structures, as well as highly specialized short life structures that are essential to successful invasion of host plants [15]. All these structural transitions are accompanied by important transcriptional and metabolic changes [16]. In addition, the ability of these organisms to develop on a wide range of plant tissues, varying in nutrient concentrations and nature, as well as artificial media, implies that they possess a great metabolic adaptive potential. The mechanisms underlying this flexibility began to be partially unveiled with the development of -omics initiatives. They were initially developed on *P. infestans* [1] and *P. nicotianae*, a soilborne pathogen able to attack aerial tissues and cause diseases on >250 plant species [17]. These analyses, mainly based on transcriptomics data, provided information on the organization of major metabolic pathways occurring in *Phytophthora*, revealing oddities and innovations. As an example, organization of the glycolytic pathway strikingly differs from what is observed in most organisms [16,18,19,20,21,22]. We first observed that a wide majority of glycolytic steps are encoded by families of 2–8 genes [18,19,20,22]. We also identified gene fusions, the replacement of ATP-dependent enzymes by pyrophosphate (PPi)-dependent ones [18,19], and the compartmentation of several enzymes to mitochondria [22].

The duplication of glycolytic genes may lead to several consequences. If they are expressed concomitantly, and in the absence of genetic drift, gene duplications may contribute to an increased rate of the glycolytic flux, as a response to specific nutritional environments [23]. Alternatively, duplicated sequences may evolve specialized functions, such as stage-specific expression, modified kinetic properties (subfunctionalization), or even different functions (neofunctionalization) if they provide a selective advantage to the organism. In the case of *Phytophthora*, the observed dual compartmentation of glycolytic enzymes (cytosol and mitochondria), after gene duplication, suggests that glucose metabolism is more complex than expected in this class of pathogens [21,22,24,25]. Whether such versatility improves the fitness of the pathogen along its life cycle is unknown. Whether it promotes its virulence also lacks information, but these questions are crucial to develop novel control strategies and improve cultural practices, such as the use of hydroponics, through a gained knowledge on the nutritional strategies of *Phytophthora* spp.

Transcriptional analyses on *P. nicotianae* led to the identification of several sequences, annotated as potential glucokinases and fructokinase of possible bacterial origin, instead of hexokinases, otherwise found in eukaryotes, an apparent novelty shared by some amitochondriate organisms, generally pathogens [26,27]. We, thus, initiated a focused analysis of this gene family that assumes the first step of glycolysis, in order to gain insights into the possible diversification of these enzymes and its possible consequences on the biology and pathogenicity of *P*. *nicotianae*. Using a combination of phylogenetic, enzymatic, and transcriptional analyses, we reveal that this multigene family has a complex, innovative evolutionary history that is restricted to the genus *Phytophthora* among oomycetes. We show that these genes encode functional, substrate-specific glucokinases with individual specificities. They are expressed differentially across the *P. nicotianae* life cycle, including plant infection. We propose that the evolution and diversification of the glucokinase gene family led to the acquisition of particular functions for its various members, as evidence of neofunctionalization. This adaptive phenomenon may have shaped some aspects of plant parasitism of *P. nicotianae*.

## 2. Materials and Methods

### 2.1. Microbial Strains and Culture Conditions

*Phytophthora nicotianae* strains are described in Appendix A. They were maintained under liquid nitrogen, within the INRAE-ISA collection, and routine cultures were conducted at 24 °C on clarified V8 medium. Alternatively, complete genomes, deposited under the name *P. nicotianae* or its synonym *P. parasitica* [17], were retrieved at GenBank. Zoospores were obtained as described [28]. Cysts were obtained by vortexing zoospore suspensions for 2 min, followed by 1-h incubation at 25 °C in water. To evaluate the ability of *P. nicotianae* to grow on different sugar sources, 1 × 10^3^ zoospores were used to inoculate media containing 1.5% agar, derived from glucose asparagine medium [29], and amended with various sugars. *Escherichia coli* DH5α and Rosetta DE3 strains (Novagen) were cultured in LB medium at 37 °C.

### 2.2. Plant Material and Infection Assays

Tomato (*Solanum Lycopersicon* cv Moneymaker, Saint Pierre, and cv Marmande) and tobacco (*Nicotiana tabacum* var Xanthii) plants were grown in vitro for 4–5 weeks at 24 °C under a 16 h light/8 h dark photoperiod for roots inoculation and in soil for leaves experimentation.

### 2.3. Detection of Hexose Kinase Activities

Detection of hexose kinase activity in *P. nicotianae* mycelia, production and expression of recombinant proteins and evaluation of enzymatic parameters are described in the Appendix A.

### 2.4. Sequence Manipulations and Phylogenetic Analyses

Hexose kinase sequences were initially identified from annotations of various *P. nicotianae* EST libraries, generated in our laboratory [18,19,30]. These nucleotide sequences were first mapped on the *P. nicotianae* INRA-310 assembly that was released at GenBank (GCF_000247585.2), then on a local PacBio-derived updated assembly. Further retrieval of glucokinases from other species was achieved through Blast (Blastp, Blastn, and TblastN) analyzes in public databases. Searches for known repetitive sequences and transposable elements were performed at Repbase using the Censor tool [31]. Prediction of subcellular localization of glucokinases was performed using Deeploc [32]. The presence of potential peroxisome targeting signal type 1 [33] was determined using the PTS1 predictor on-line tool (https://mendel.imp.ac.at/pts1/ accessed on 28 September 2021), while the PTS2 was determined by manual inspection of sequences and comparison to the PTS2 consensus sequence [34]. Prediction of transmembrane domains was performed using the TMHMM server (http://www.cbs.dtu.dk/services/TMHMM; accessed on 28 September 2021). Secondary and 3D structures were predicted using Phyre2 [35] and the iterative threading assembly refinement server I-TASSER [36]. Energy minimization was performed using ModRefiner [37]. Structures were further aligned using FATCAT [38] for pairwise comparisons and POSA for multiple alignments [39]. Rendering of models was performed using ChimeraX [40]. Pairwise comparisons were performed with LALIGN (https://embnet.vital-it.ch; accessed on 14 June 2021). Protein sequences alignments were made using MUSCLE [41] in the SeaView v4.0 software [42]. Phylogenetic trees were constructed by maximum likelihood (ML), using the best fitting substitution model that was LG, with a discrete Gamma distribution (+G) and five rate categories, using MEGA [43]. Branches were supported by 100 bootstrap replicates. Trees were designed using the Interactive Tree Of Life (ITOL) online tool [44].

### 2.5. RNA Isolation and Transcriptional Analyses

Total RNA from individual stages of the *P. nicotianae* life cycle, including tomato (leaf and root) and tobacco (leaf) invasion, was obtained according to Laroche-Raynal et al. [45]. RNA (1 µg) was initially treated with DNAse I (Ambion), and cDNAs were synthesized using the SuperScript^®^ IV Reverse Transcriptase (Invitrogen). Real time qPCR was performed with 5 µL of cDNA (diluted 20-fold), using Brilliant III Ultra-Fast SYBR (Agilent Technologies, Santa Clara, CA, USA). The *P. nicotianae* gene encoding Ubiquitin conjugating enzyme was used as the constitutive standard for normalization of transcripts, using the primers described in Appendix A. Normalization was performed according to Vandesompele et al. [46]. Gene-specific oligonucleotides were designed using PRIMER3 software (http://frodo.wi.mit.edu; accessed on 14 September 2020), and their specificity was validated by the analysis of dissociation curves.

## 3. Results

### 3.1. Phytophthora—Specific Amplification of Glucokinase Genes among Oomycetes

Mining the genome of *P. nicotianae* INRA-310 with a set of query sequences extracted from several *P. nicotianae* cDNA libraries [18,19,30] revealed four genes, annotated as glucokinases (PPTG_18927, PPTG_18933, PPTG_18934, and PPTG_18935). The predicted proteins possessed a molecular mass (38.65, 38.56, 38.59, and 43.64 kDa, respectively) that diverged from eukaryotic broad-substrate hexokinases [47], but that were similar to those calculated for prokaryotic, glucose-specific glucokinases [48,49]. Sequence alignment indicated that they were highly conserved and constituted three protein types that were designated as GKI-III (Appendix A). GKI (PPTG_18934) possessed a PTS2 [34] motif, suggesting that it is potentially targeted to peroxisomes. PPTG_18927 and PPTG_18933 were near identical (99.7% identity at the amino acid level, Appendix A) and collectively designated as GKII. Last, GKIII (PPTG_18935) contained a transmembrane domain at its C-terminus. No sequence related to typical, eukaryotic hexokinase, was retrieved from the *P. nicotianae* genome, whatever the developed method. In addition to these four sequences, we identified a single gene (PPTG_18886) encoding a 294-aa protein that possessed the typical features of fructokinases, which are the PFAM domain IPR000600 and a N-terminal motif (EXGXT) corresponding to the ATP-binding region [49]. It was, thus, considered a potential fructokinase and called FKI (Appendix A).

The presence of several GK genes in *P. nicotianae* was already known from the annotations of EST libraries [18,19,30] and is in accordance with a report that described up to seven GK genes in *P. infestans* [20]. We, thus, searched eventual, additional GK genes in an updated assembly of the *P. nicotianae* genome, based on long reads. We retrieved the four GK genes already identified, which were clustered within a 26.70 kb region (Appendix A). No other hit was obtained, and attempts to find additional GK genes or even traces of pseudogenes in the genomes of *P. nicotianae* publicly available under this name or under the *P. nicotianae* denomination [17] were also unsuccessful. We, thus, concluded that the *P. nicotianae* GK repertoire consists of one GKI, two GKII, and one GKIII gene. The genomic organization of these genes is conserved among *Phytophthora* spp., as illustrated in Appendix A, with the description of the GK locus in the genome of *P. infestans*. The GK genes are clustered across a 27–30-kb region, with the GKII and GKIII sequences constituting the 5′ and 3′ boundaries, respectively. The GKII cluster is interrupted by a ~20-kb region that mainly hosted sequences derived from transposable elements (Appendix A).

We searched the analogs of each class of GK, in a range of stramenopiles, to establish phylogenetic relationships between these sequences. On the course of sequence retrieval, we found several incomplete sequences that might either correspond to pseudogenes or result from incorrect gene calling. These sequences were first discarded from the global alignment that only contained entire sequences or sequences that were manually corrected. The resulting tree resolved the dataset into five main clades that were supported by high bootstrap values (Figure 1).

A first clade (clade A, Figure 1) encompassed sequences from diatoms and brown algae, which all possessed a single gene. The four other clades contained sequences from oomycetes. They followed the current taxonomy, as sequences from the Saprolegnialean lineage constituted a single clade (clade B, Figure 1). This clade was further resolved into two distinct groups that separated *Aphanomyces* sequences from a *Achlya*-*Saprolegnia*-*Thraustotheca* assemblage, in accordance with previous studies conducted using rDNA-derived markers [50]. They all possessed a transmembrane domain at their C-terminal end, as well as sequences from diatoms and brown algae. The third clade (clade C) contained sequences from Albuginales and Peronosporales, including GKIII. It was further resolved into two subclades, one containing sequences from Peronosporaceae (*Phytophthora* and downy mildews) and the other containing sequences from Albuginales and Pythiaceae. Sequences from this latter lineage were ultimately separated, according to the phylogenetic relationships of their relative species [51]. Here, again, all sequences of the clade C possessed a transmembrane domain at their C-terminal end, which appeared to be a feature conserved among stramenopiles. Another common feature was the presence of two introns of 59 and 88 bp, respectively, which interrupted the ORFs at the Ser_2_ and Asp_68_ residues of all sequences analyzed. Last, GKI and GKII sequences constituted two distinct groups (designated as clades D and E, Figure 1). We noted that GKI was present in *Phytophthora* and *Peronospora effusa*, while GKII-related sequences were found only in *Phytophthora*. We then incorporated incomplete sequences and pseudogenes in the global alignment. All of them took place in the clade E, which contained GKII sequences (Appendix A). We, therefore, proposed that GKIII is the closest relative of the original glucokinase common to stramenopiles, which suffered several rounds of duplication. A first amplification event, occurring in some Peronosporales, resulted in the adjacent GKI gene. Then, additional amplification events took place in the *Phytophthora* ancestor, with the emergence of the GKII clade.

### 3.2. Different Evolutionary Histories Shaped the P. nicotianae GK Subclasses

To further evaluate the relationships underlying GK evolution, we mined *P. nicotianae* genome sequences available at GenBank or generated on the course of sequencing projects in our lab. Alternatively, we also directly sequenced GK transcripts from few isolates (Appendix A). The quality of sequences diverged among classes. Hence, the GKIII genes were easily retrieved from the various genomes. Conversely, the GKI and GKII members were frequently located on boundaries of their respective contigs and scaffolds, so that only partial sequences could be resolved. They were generally retrieved in short-read derived genome assemblies, so the repetitive nature of these sequences can make them refractory to assembly [52].

We retrieved 13 complete GKI homologs. They corresponded to intronless sequences, and the global 1066-bp alignment revealed 38 variable sites. Yet, most of them were synonymous, and we only identified four amino acid changes (Appendix A). Six genes shared two regions accumulating synonymous mutations, corresponding to codon changes on two close, seven-residue stretches (Appendix A). These genes originated from *P. nicotianae* isolates, primarily collected on plants other than tobacco (Appendix A). A phylogenetic analysis of GKI nucleotide sequences confirmed the grouping of tobacco isolates, and further resolved discrete associations related to host of origin, exemplified by the three tomato isolates or two citrus isolates (Figure 2A). The tree further separated tobacco isolates, collected in Chinese fields, from those originating from other locations (Figure 2A).

A total of 33 GKII members were retrieved from 14 *P. nicotianae* isolates, but only 19 corresponded to complete ORFs and were from 5 tobacco strains and 8 isolates from other hosts. They were quite equally distributed between GKIIa and GKIIb members. We identified 28 variable sites on a 1060-bp global alignment (Appendix A). They consisted of 13 synonymous substitutions and 15 mutations, leading to 7 amino acid replacements and the introduction of a stop codon. Most non-synonymous mutations were located in the 3′ end of the coding sequences (Appendix A) and constituted the main contributors to the differentiation between GKIIa (illustrated by PPTG_18927) and GKIIb (PPTG_18933, Appendix A). A phylogenetic reconstruction of GKII sequences did not reveal a topology associated with the geographical or plant origin of the isolates, and no evidence of GKIIa and GKIIb as founder genes could be drawn (Figure 2B).

Last, 16 sequences were ascribed to GKIII. A global 1346-bp alignment included the entire ORF, interrupted by two introns of 59 and 88 bp, respectively. The position, length, and nature introns were deduced from the mapping of cDNAs to the genome sequences. Only six differences were scored, of which four occurred only once and might correspond to sequencing errors, as well. Three substitutions were located into introns, the latter mutations consisting of one synonymous substitution and two non-synonymous substitutions (E175Q and A387V, respectively). Given this level of extreme conservation, no phylogenetic analysis was performed on this set of sequences.

We noted that the two stretches of amino acids, harbored by GKI sequences, whose codon usage discriminated tobacco and non-tobacco isolates, were highly conserved among the four GK genes among oomycetes (Appendix A). So, the nucleotide sequences corresponding to these two blocks were retrieved from the *P. nicotianae* dataset and assembled in a concatenated alignment (Appendix A). Only four GKIII sequences were included in the alignment because they are strictly conserved at these loci. The resulting tree separated the GKI dataset on two distinct branches, one consisting of sequences from non-tobacco isolates, while the sequences from tobacco isolates were grouped in a robust clade with GKII sequences, irrespective of the host plant. Last, GKIII sequences were clearly isolated. This observation supported our previous results, indicating that different forces drive the evolution of *P. nicotianae* GK genes. It further reveals the unexpected connection between GKI and GKII sequences from tobacco isolates, suggesting possible recombination events in the central part of these genes, without consequences on the structures of the corresponding proteins, as these possible swaps involve only synonymous mutations.

### 3.3. P. nicotianae GKs Are Substrate-Specific Enzymes with Distinct Properties

We further investigated the structural features of potential GKs. We first deduced the secondary structures of the proteins, which were predicted to mainly consist of 11–15 alpha-helices and 14–15 beta-strands (Figure 3A), in the range of what was determined for the *E. coli* glucokinase [53]. Overall, the predicted structures were highly similar, but local differences were expected to potentially impact the tertiary structure of each protein (Figure 3A). We took advantage of the availability of crystal structures for several glucokinases for modelling the three-dimensional structure of the *P. nicotianae* GKs, using two independent methods. Using Phyre2, models were generated with 100% confidence and 83–98% coverage (Appendix A). The most similar structures displayed >20% identity with the various GK members and corresponded to crystal structures of glucokinases from *Naegleria fowleri* (PDB:c6da0A), antarctic psychrotroph *Streptomyces* (PDB: c3vpzA), and *E. coli* (PDB: 1sz2A, Appendix A). We also used I-TASSER, which generated five possible 3D structural models [36]. For all sequences, I-TASSER identified the three same structures (Appendix A). The diverse parameters, such as confidence score (C score) and template modelling (TM) score, associated with the high coverage percentage, indicated the reliability of the modeling and a correct topology (Appendix A). So, the predicted GK were unambiguously modeled against glucokinases. Pairwise alignments were performed with the three templates and three *Phytophthora* predicted structures. We noted differences in ranking between the three models that revealed preferred associations for each GK class (Appendix A). So, the structures were superposed to their respective top template (Figure 3B). Overall, the three structures were closely related and exhibited a conserved topology matching known structures of glucokinases. Yet, superposition of the three *Phytophthora* predicted models revealed several differences, especially with the presence of a helix specific to GKII in the 227–240 region of the protein or the C-terminal extension found on GKIII, which contained the potential transmembrane domain (Appendix A). We could, thus, anticipate functional divergence among the various *P. nicotianae* GKs, based on enzymatic properties and structural specificities.

Bacterial glucokinases generally display a strong specificity towards glucose, unlike eukaryotic hexokinases, which are rather broad-substrate enzymes [47,48]. We, thus, intended to determine whether *P. nicotianae* glucokinase candidates, which display features typical of prokaryotic enzymes, are specific for glucose. In the affirmative, fructokinase was expected to display some specificity for fructose.

We first studied the ability of *P. nicotianae* to grow on different sugar sources. Growth was observed on glucose, fructose, and, to a lower extent, mannose, as well as sucrose and maltose, which generate glucose and/or fructose upon degradation. In contrast, *P. nicotianae* growth was impaired using other hexoses, even on trehalose, although composed of two glucose molecules and alternate carbon sources (Appendix A). *P. nicotianae* also failed to grow on some gluconeogenic carbon sources, such as acetate or glycerol.

We characterized hexose kinase activity in crude extracts from *P. nicotianae* after electrophoresis in acrylamide gels under non-denaturing conditions [55]. Using either glucose or fructose in the staining mixture indicated that glucokinase and fructokinase activities are carried out by separate proteins (Appendix A). Substrate specificity of *P. nicotianae* hexose kinases was further assessed using recombinant proteins. To this aim, the entire ORFs of GKI, GKII, FKI, and the ORF of GKIII deleted from the transmembrane domain were expressed in *E. coli*. The recombinant proteins were purified to near homogeneity and characterized, with respect to their biochemical properties and substrate affinities. Kinetic parameters were determined according to the Hanes-Woolf procedure and summarized in Table 1.

The potential glucokinases tested here are active enzymes. Their *Km* values (40–125 μM) revealed high affinity towards glucose, and they are not inhibited by glucose-6-phosphate. The GKI and GKII recombinant proteins displayed a *Km* for glucose that was approximately three to four times lower than GKIII, indicating a strong increase in affinity for the sugar. Conversely, their respective *Km* for ATP was two to three times higher than the calculated *Km* of GKIII. No enzyme was able to use PPi, and they are unable to phosphorylate fructose. GKI (but not GKII or GKIII) is inhibited at glucose concentrations of ≥1 mM. ATP is the sole phosphoryl donor for GKI and GKII, while GKIII can use various nucleotides, albeit with a lower activity. PPTG_18886 encodes a specialized fructokinase, which displays high affinity towards this sugar and is unable to phosphorylate glucose. These results confirmed that the hexose kinases of *Phytophthora* are specialized enzymes, consisting of specialized glucokinases (GK, E.C.2.7.1.2) and fructokinase (E.C.2.7.1.4). We could also conclude that, despite a very high sequence conservation, the three types of GKs display differences enough to ascribe them potential different functions in *Phytophthora* biology.

### 3.4. Phytophthora GKs Have an Ancient Origin

As the *Phytophthora* GK accumulate commonalties with bacterial enzymes, a possibility is that they have been acquired by one or several horizontal gene transfer (HGT) events. HGT can be defined as the transmission of genes between organisms by other ways than direct inheritance from parental lineages to their offspring [56]. A broader definition is the transfer of genes between reproductively isolated genomes [57]. HGT appears to be an active driver of eukaryote evolution, especially in the acquisition of pathogenic traits and adaptation of microbes to their environment [58]. The multiplication of sequencing projects favored the emergence of comparative studies that highlighted the role of horizontal gene transfer (HGT) in the evolution of oomycetes [57,58,59,60]. More precisely, ~50 gene families were proposed to be transferred into oomycete genomes, the *Phytophthora* lineage being subject to this phenomenon at a higher frequency [58]. A large majority concerns genes encoding enzymes involved in carbohydrate metabolism that were likely acquired from fungi, but cases of prokaryote-to-oomycete transfers have been suspected [60]. So, we performed additional investigations on the possible origin of the four *P. nicotianae* GKs. We conducted Blastp searches against NR at Genbank and retained the 5000 best target sequences. We then discarded sequences from *Phytophthora* and analyzed the origin of hits. As a rule, we did not retrieve any sequence from fungi, animals, or land plants in this 5000-hit list. Rather, the *Phytophthora* GKs matched with sequences from other stramenopiles, including diatoms and brown algae, as expected, but also with few lower eukaryotes belonging to various ‘supergroups’ that constitute the current eukaryote tree of life [3]. Additional Tblastn searches gave similar results. We, thus, selected representatives from each organism identified in the blast search and explored their phylogenetic affinities. Although absent from the first analysis, potential glucokinases originating from fungi and metazoans were retrieved on the course of a keyword-based search and included in the analysis. The phylogenetic tree clearly resolved all oomycete sequences in a single robust clade (Figure 4). A higher level grouped them with other eukaryotic sequences belonging to the stramenopiles and alveolates (the dinoflagellate *Symbiodinium natans*), as well as to distant groups, such as rhodophytes, haptophytes, and unicellular relatives of opisthokonts within Obazoa [61]. Prokaryotic sequences from cyanobacteria and proteobacteria were confidently separated from this eukaryotic assemblage. Glucokinases from trypanosomatids and amitochondriate protists, all belonging to the Excavata group [3], were isolated from the rest of the sequences. Last, sequences retrieved through keyword-based searches were tightly grouped into a distant, apparently unrelated clade that corresponded to hexokinases. Another observation was that, although grouped within an ‘eukaryotic branch’, GKs from oomycetes were separated from those of other stramenopiles, while diatoms and brown algae were represented within a single clade (Figure 4). In addition, the patchy distribution of GKs within the eukaryotic assemblage reveals a complex evolutionary history of these sequences. As a rule, the *Phytophthora* GKs have an ancient origin, but do not derive from fungi, as do other genes of importance, such as carbohydrate-degrading enzymes [57,59]. Additionally, there is no evidence of a bacterial origin to GKs, despite shared characteristics. So, if they have actually been acquired from bacteria, phylogenetic analyses indicated that such event(s) should have occurred very early and are no longer tractable.

### 3.5. Transcriptional Profiles Support Neofunctionalization of GK Genes along P. nicotianae Life Cycle

To evaluate whether the GK genes have been adapted to diverse functions, as suggested by the various experimentations presented above, we established their transcriptional profile by qRT-PCR. We first evaluated the relative expression of each gene in pre-infection structures, including swimming zoospores, cysts, and germinating cysts. Transcription profiles were performed on six strains, isolated from tobacco or tomato. As a rule, we could not observe any bias in the expression levels possibly associated to the strain and its origin. Yet, we noted important differences in the relative expression of GKs, whatever the strain and physiological stage analyzed (Figure 4). Hence, GKI was by far expressed at the highest levels, while GKIII expression was barely detectable (Figure 5A). Refined analysis revealed important discrepancies in expression profiles across the physiological stages. Hence, GKI is markedly more expressed at unicellular stages, i.e., motile zoospores and cysts, than in germinating cysts (Figure 5B). Conversely, GKII expression is very low at unicellular stages and dramatically increased in germinating cysts. Last, the overall expression of GKIII was extremely low and decreased only slightly in germinating cysts (Figure 5B). We, thus, concluded that, in addition to differences in sequence, potential structure, evolutionary trajectories, and enzymatic properties, GKs from *P nicotianae* also diverged by their expression profile. So, GKI appeared to be highly associated with the unicellular stage, and GKII was rather associated to germinating cysts. GKIII apparently displayed a transcriptional pattern similar to GKI but at a strikingly lower level (Figure 5B).

We then estimated the relative expression of GK genes during plant infection. To this aim, several tomato and tobacco plants were inoculated with zoospores of several strains and sampled at two-, four-, and six-days post inoculation (dpi). The experiment included a total of 13 host-pathogen associations. We noted a strong heterogeneity among expression values (Figure 6A). It was particularly visible for GKI and GKII (Figure 6B,C), as the amplitude of expression values varied from 1 to 10 among strains. On the opposite, GKIII was expressed at similar, very faint levels in all cases, compared to the other GK genes (Figure 6D). Yet, a main result was that GKII expression was strikingly increased, compared to the values scored in pre-infection structures, in frame with the high induction observed in germinating cysts (Figure 5). So, expression of GKII could be associated to the infection step. We could also note that the expression levels of each GK gene, relative to the constitutive control, did not diverge along the kinetics.

Attempts to eventually observe, through cumulative analyses, an effect of host variety on the overall expression level of GK transcripts were unsuccessful, at the noticeable exception of the tobacco assay, where GKIII was expressed at very high levels, compared to its relative expression in tomato roots (Appendix A). So, the important variation, observed in the expression values of GKI and GKII, could be related to the variability of pathogenicity of the various strains and subsequent differences in the extent of tissue invasion. This observation prompted us to investigate the possible relationships between GK expression and components of pathogenicity of *P. nicotianae*. We inoculated tomato leaves with zoospores of *P. nicotianae*, as this soilborne pathogen is also able to successfully invade aerial tissues [63], and that tissue invasion can be easily monitored and quantified more efficiently than on roots. Twelve host-pathogen associations were analyzed, using three tomato varieties and four *P. nicotianae* isolates. Leaves were totally invaded at 4 dpi, so expression of GK genes and measurement of leaf invasion were assessed at 3 dpi (Figure 7). As a rule, GKI and GKIII relative expression was similar to what was observed in root infection. Yet, we noted a striking decrease of GKII expression, relative to the constitutive UBC gene (Figure 7A). As performed with the root assay, we investigated the expression of GK genes at the variety level. Results were quite similar to before for GKI and GKIII, whose expression was not modified among infected varieties (Figure 7B). However, GKII expression was markedly different from one variety to another. So, leaf infection was accompanied by a stronger expression of GKII in Saint Pierre, compared to the two other varieties (Figure 7B).

Expression of GK genes was also investigated to eventually characterize differences linked to the nature of the *P. nicotianae* strain. We detected differential results. Hence, the expression of the three GK genes was significantly lower when examining *P. nicotianae* 149 than what was scored in another strain (Figure 8). Yet, we could not establish a clear ranking that would be identical in all cases. Expression of GKII and GKIII was higher in the case of the strain 709, compared to 149 (Figure 8, middle and 8, right), but no significant difference was observed for GKI. In conclusion, we observed few changes in GK expression that were supported by statistical analyses in the different host-pathogen associations. Yet, expression values still presented a high amplitude, which we intended to correlate to an eventual extent of invasion.

We, thus, evaluated the extent of tissue invasion in each association. This was achieved by monitoring the expression of a tomato gene, relative to the expression of a *P. nicotianae* constitutive gene (Figure 9, left), and direct measurement of the surface of invaded leaf tissues (Figure 9, right). The expression of the tomato gene PSKR was monitored to reflect the extent of living tissues. Its expression level was relatively high in tomato leaves infected by the strain 149, compared to its expression in other infection assays, in particular in infections by strains 709 and 721 (Figure 9, left). Markedly, the extent of leaf invasion was the lowest in the infection by 149, compared to the surface of invasion by other strains, especially with the strain 709 (Figure 9A, right). We could then conclude that 149 was less aggressive in the conditions of our assay than the other strains, while 709 was significantly more aggressive. This result was compared to the data concerning the expression of the GK genes presented in Figure 8. Empirically, we could define a correlation between the level of GKII expression and aggressiveness of the *P. nicotianae* strains. It was particularly obvious in the case of 149 and 709. This intuitive interpretation was further supported by a Kandal’s tau test, which brought several conclusions (Figure 9B). First, the expression of GKI was not correlated to leaf invasion or the expression of the other GK genes. In contrast, GKII expression was moderately correlated to the expression of GKIII and correlated to the extent of leaf invasion. Last, the expression of the tomato PSK was negatively correlated to the extent of infected area, as expected. This last experiment confirmed the differential expression of the three GK genes and further identified a potential involvement of GKII, but not the other genes, in the intensity of infection symptoms and, consequently, to the level of aggressiveness of the *P. nicotianae* strains.

## 4. Discussion

The organization of glycolytic genes into small multigene families in *Phytophthora* could either correspond to the need of high glycolytic flux towards energy production or an evolutionary process that would eventually promote functional diversification. Addressing this question is crucial for a better understanding of the mechanisms underlying the metabolism of these major plant pathogens, as a first step towards defining innovative crop strategies. We confirm, experimentally, in the present study that *Phytophthora* spp., exemplified by *P. nicotianae*, possess a set of glucokinases (GKs) and fructokinase (FK), instead of a typical hexokinase (HK), to enter glycolysis. Here, three highly conserved, although distinct, types of proteins can phosphorylate glucose, but not other sugars, while the single fructokinase activates only fructose in vitro. Focusing on glucokinases, we show that they are encoded by a small gene cluster, driven by a complex evolutionary scenario. Moreover, we identified distinct sequence, structure, enzymatic, transcriptional, and evolutionary patterns that suggest functional divergence and possible neofunctionalization of this multigene family. More important, expression of one, but not all, GK genes is correlated to pathogenicity.

It was previously proposed that *Phytophthora* acquired glucokinases via a horizontal transfer event [18,22]. Hence, GKs share several features with glucokinases of prokaryotes, such as their length or enzymatic properties and specificity towards glucose. This is in frame with the general observation that sugar-specific kinases are a characteristic of prokaryotes and unicellular organisms, while multicellular eukaryotes rather possess hexokinases with a broad substrate specificity [47]. Yet, the *Phytophthora* GKs differ from typical glucokinases by at least two features. First, we show that the phylogenetic affinities with the bacterial sequences that were proposed previously are not as valid as stated before and must be reconsidered. GKs from *Phytophthora* are nested within a robust clade of sequences from stramenopiles and other lower eukaryotes and are distant from bacterial sequences. So, if they originate from a bacterial donor, this acquisition occurred very early in the evolution of the stramenopiles group. *Phytophthora* GKs may, thus, be considered “oomycete-specific” proteins. Second, glucokinases generally display significantly lower (up to 50-fold) affinity for glucose than hexokinases, as shown by *Km* values that are generally in the millimolar range [47,65]. With *Kms* for glucose ranging between 40 and 125 µM, the *P. nicotianae* GKs have properties close to plant hexokinases that generally display a 25–150 µmM range [66]. On the other hand, they share the insensitivity to G6P with bacterial GKs, which otherwise inhibits HK activity [47]. Similarly, the *Km* of hexokinases for fructose is generally in the millimolar range, while the calculated *Km* of the *P. nicotianae* FKI is still in the micromolar range. This highlights the particularly high affinity of *Phytophthora* FK and GKs for their respective substates, and a deviation from glucokinases of bacteria and unicellular eukaryotes. This reinforces the idea that, if these enzymes have been acquired laterally from a bacterial donor, this event took place very early, so that these genes and their associated function have been modelled by *Phytophthora* on the course of evolution.

A widely accepted view is that ancestral enzymes were generalists and that modern, specialized proteins evolved after gene duplication and refining their substrate recognition [67,68]. On the contrary, the specificity of hexose kinases towards various sugars is observed in prokaryotes and unicellular organisms, as well as in *P. nicotianae* (this study), while non-specific hexokinases are typical of higher eukaryotes [47]. Versatility towards a range of substrates may be proposed as an advanced, opportunistic character that organisms developed to avoid subordination to a single substrate, as well as to benefit of all compounds available, eventually to the prejudice of catalytic efficiency [67]. In the present case, the substrate specificity of GKs and FK has been determined in vitro, using standard conditions. So, we cannot rule out that both GKs and FK do not have an absolute specificity; rather, they possess some level of substrate promiscuity under normal conditions [69,70]. Another explanation is directly linked to the *Phytophthora* lifestyle. We show here that *P. nicotianae* grows on only few sugars (mainly glucose and fructose, and their precursor, sucrose) and, thus, does not require a broad-substrate enzyme to initiate glycolysis and other metabolic pathways involving glucose-phosphate. In addition, sucrose is the major product of photosynthesis in plants; its degradation is achieved by sucrose synthase and invertases, yielding fructose and UDP-glucose or fructose and glucose, respectively [71]. Consequently, they represent the main sugars available for uptake by *Phytophthora* during plant infection. It is also well-documented that the activation of invertases and subsequent increase in glucose and fructose availability is a component of defense responses upon pathogen infection [71]. Furthermore, *Phytophthora* possess several invertases, which are also expressed during the biotrophic steps of plant infection and secreted inside haustoria [72]. In this context, simultaneously diverting both the cleavage products of plant sucrose for its own metabolism using high affinity, sugar-specific kinases is a metabolic advantage for *Phytophthora*, in that it avoids an eventual competition between two substrates for a single enzyme. The development or acquisition of substrate-specific hexose kinases in *Phytophthora* would then result from a co-evolutionary mechanism facing its biotic environment and may not considered a rudimentary trait. *Phytophthora* genomes contain 4–8 GK genes, belonging to three sub-clades and organized in small clusters that display a highly conserved organization and similarly harbor pieces of transposable elements at conserved locations. Thus, the generation of the GK cluster and its invasion by transposable elements is likely to have predated species radiation. Surprisingly, this innovation is observed only in *Phytophthora*, as more “primitive” oomycetes, such as Saprolegniales, or more “evolved” species, such as obligate pathogens, only possess GKIII. All organisms belonging to stramenopiles analyzed in this study also possess a single glucokinase gene, which, based on sequence alignments, the presence of a C-terminal transmembrane domain in the predicted protein and a similar exon-intron structure is akin to GKIII that consequently may reflect an ancestral form of GK and the likely founder of the subsequent families. The potential advantage of the *Phytophthora* species is to possess several GK enzymes, whereas all other oomycetes, and to a wider extent, stramenopiles, investigated so far harbor only one gene, which constituted an important question of the present study. The acquisition of multiple glucokinase genes, through successive duplications by an ancestor of Peronosporales and Albuginales, and the subsequent loss of most of them in all organisms but *Phytophthora*, appears unlikely. The number of GKII copies varies from one species to another, possibly resulting from an asynchronous duplication process occurring during the evolution of *Phytophthora* species. Yet, most of them consist of pseudogenes, so that only one or two GKII genes are potentially expressed. Beyond the mechanisms driving the genesis of the GK family in *Phytophthora*, we also have to question why they retained such complexity. It may be argued that the *Phytophthora* species have a more versatile lifestyle than other oomycetes, as they are able to infect a wide diversity of hosts and tissues, in addition to rest in soil and grow on artificial media [21]. Consequently, GKI and GKII might exert functions that are either dispensable in other oomycetes or assumed by other proteins.

Identifying additional or alternative functions to GKs is a challenging task because they retained a canonical phosphorylating activity on glucose. Yet, we may speculate on eventual individual function of each GK class. The slight differences observed at the sequence level turned out to impact the potential conformation of GKs. The 3D structure of the three GKs were modelled by two methods. The reliability and accuracy of the prediction models were comforted by high-quality indexes. Hence, TM-scores > 0.5 indicated correct topologies of the models, and C scores were ≥ −1.0, so that the models were highly reliable. Using such criteria, we showed that each class can be associated to a given structural template and, thus, may possess subtle structural differences. Whether they contribute to interactions with specific ligands, in addition to glucose and ATP, or even proteins, is unknown, and interactor screens would maybe reveal unexpected properties of each GK class. GKI and GKII have a lower affinity for ATP, their unique phosphate donor, than GKIII, although they apparently adapted for a better affinity towards glucose. In addition, GKI is apparently inhibited by high glucose concentrations. So, they may have evolved to be sensitive to the glucose concentration, towards a possible role of sensor, while GKIII would fulfill a basal glycolytic activity, even using a range of phosphate donors, as a suggestion of opportunistic behavior.

Predicting GKs various subcellular localizations for GKs may also help towards elucidation of their actual functions. Hence, GKI would be targeted in a peroxisomal or assimilated structure, while GKIII was predicted to be associated to membranes, GKII being likely located into cytoplasm. The predicted location of GKI to peroxisomes is in frame with other observations. In Trypanosomatids, glycolysis is known to take place in glycosomes, peroxisome-like organelles, along with other metabolic pathways [73,74]. A possible peroxisomal location of several glycolytic enzymes has been also proposed in fungi, resulting from stop codon read-through and alternative splicing, which uncover a cryptic peroxisome targeting signature (PTS) motif in the corresponding proteins [75]. Motif searches in the predicted proteome of *Phytophthora* and location investigations using fluorescent probes are required to confirm whether peroxisome-like organelles occur in *Phytophthora* and if they host glycolytic enzymes. Partitioning of enzymes ensuring glycolysis, an otherwise cytoplasm-located metabolic pathway, has been largely documented in stramenopiles. Especially, the lower part (pay-off stage) of glycolysis, which involves three-carbon intermediates, takes place into mitochondria of these organisms [22,24,25]. Glycolytic enzymes have also been found in the mitochondrial fractions of *Arabidopsis* and human cells during proteomic analyses [76,77]. Such location would optimize the channeling of pyruvate, the end-product of glycolysis, towards the mitochondria, where it is used as a substrate for respiration. If GKIII is actually located at the mitochondrial membrane, it might constitute a metabolic advantage, as its association with the mitochondria-located, ATP-generating steps of the glycolysis would provide a favored access to ATP. In addition, it would concentrate enzymes and their associated metabolites in a reduced environment, which is important in the case of coenocytic organisms, such as *Phytophthora*. Keeping in mind that most glycolytic enzymes are encoded by multigene families and the corresponding proteins are predicted to be located either in the cytoplasm or in mitochondria, it is also possible that glycolysis, or at least some steps, may occur concomitantly in several locations.

GKI clearly has a fundamental role in the physiology of *Phytophthora* because of its high expression, especially in the unicellular structures, the selection pressure acting on this single-copy gene, as well as the apparent lack of connection between its expression level and *P. nicotianae* virulence. We show here that GKI genes present a sequence polymorphism, associated with host specificity. Hence, we could discriminate strains isolated on tobacco from those collected on other hosts. We could further distinguish tobacco strains from China from isolates from other countries. Subsequently, we could also identify tomato and citrus isolates, as well as strains from ornamentals. Molecular bases of host specificity in *P. nicotianae* have been investigated for decades using a range of molecular tools [78,79,80], and the occurrence of cryptic species displaying host specificity in a *P. nicotianae*/*P. nicotianae* species complex, displaying a broad host range, has been frequently evoked [17,80]. So, the GKI genes may constitute adequate diagnostic tools. Surprisingly, the polymorphism characterized in the present study mainly consists of silent mutations, so that the correlation with host range is not obvious. GKI expression is highly associated with unicellular structures, such as motile zoospores, which constitute the main source of inoculum and cysts, which result from the loss of flagella and the subsequent formation of a thick cell wall [81]. Flagella and cysts are the earliest *Phytophthora* structures in contact with the host plant before germination and the development of coenocytic hyphae. It is known that these different stages possess distinct, specific transcriptional programs [16,82,83,84]. We can, thus, suppose that unicellular forms possess a set of tRNAs that differ from what is found in the invading hyphae and that this tRNA distribution is crucial for the successful development of the genetic program accompanying the very early steps of infection. We can also suppose that this feature is under high selection pressure of the plant host. Interestingly, discriminating nucleotides harbored by tobacco isolates are also present in GKII genes. As GKII is likely to have originated from the amplification of GKI, we can suppose that host specialization occurred after the achievement of the amplification of the GK family.

The levels of GKII mRNA are very low in infected leaves, where glucose is abundant, but very high, compared to preinfection structures and invaded leaves in infected roots, which are a typical sink organ [85]. The observed correlation between its expression level in planta and the extent of tissue invasion suggests that GKII may have evolved original functions linked to virulence. Last, GKIII expression is very low, but almost unchanged, whatever the situation analyzed, while GKI and GKII display noticeable changes. We can, thus, propose a hypothetical scheme ascribing the different GKs a distinct role. GKI would be associated with unicellular forms of *Phytophthora*, GKII would contribute to the infection process, and GKIII would retain a basic glycolytic function. This would constitute a case of functional divergence, if not the neofunctionalization of GK genes. Such phenomenon is frequently observed in multigene families, especially those encoding metabolic enzymes [86]. Duplication events produce gene copies, which may be redundant to the original one. Then, the founder gene may retain its function, while duplicate genes progressively accumulate mutations, until losing the original enzymatic properties or disappearing if mutations are deleterious. In the present case, mutations are limited, as the overall sequence conservation is extremely high and all tested sequences retained a glucokinase activity.

If GKII evolved original functions linked to virulence, as deduced from the correlation between its expression level in planta and the extent of tissue invasion, its absence in other Peronosporales suggests that this gene is important, but non-essential, for the pathogen. This hypothesis is supported by the observation that only GKII suffers gene amplification and pseudogenization events that do not target GKI and GKIII or have lethal effects. The precise contribution of GKII to the infection process is unknown. GKII, with an apparent high affinity for glucose, would act as an intracellular sensor, if not a transporter, which would deliver glucose molecules to their final destinations before being integrated to various processes. It would act in conjunction with secreted invertases that have been proposed to provide *Phytophthora* with glucose molecules generated from the breakage of plant sucrose molecules [72]. Yet, the fate of glucose 6-phophate (G6P), the product of the reaction catalyzed by GK, goes far beyond the sole glycolysis and energy production. Among the various metabolic pathways involving G6P, we can identify protein glycosylation, synthesis of several amino acids or pentose phosphate pathways (PPP), although several connections occur between these latter pathways. The importance of this pathway has been neglected for long time. Yet, PPP contributes, in addition to glucose catabolism, to synthesis of nucleotides, amino acids, fatty acids, and vitamins [87]. All these metabolites are obviously necessary for *Phytophthora* development during plant infection. Hence, amino acid biosynthesis is highly increased during plant infection by *Phytophthora* [88,89], reinforcing the possible contribution of GKII to this phenomenon. PPP is also a main actor in the generation of NADPH [87]. So, GKII would contribute to *Phytophthora* metabolism during in planta growth and might have a protective effect against oxidative stress during plant infection, due to its commitment to increase NADPH levels for scavenging plant-defense-associated reactive oxygen species. Monitoring the pentose phosphate pathway during plant infection is necessary to validate this hypothesis. Yet, whatever the function(s) assumed by GKII, the present study clearly shows that this class of genes evolved from an ‘ancestral’ sequence; with its expression being linked to plant infection, it actively contributes to parasitism, as the consequence of a neofunctionalization process.

The occurrence of several hexose kinases is common among living organisms. Hence, higher plants and fungi generally possess both hexokinases and glucokinases, where they catalyze the first step of glucose metabolism but also play a major role in glucose sensing signal transduction, development, growth, cell death, and hormone pathways [90,91,92], with distinct enzymatic properties and even separate functions [93,94,95,96]. In these cases, glucokinases may even be devoid of catalytic activity, whereas the canonical hexose activity is mediated by the hexokinase. In addition, non-catalytic hexokinases have been found in many genomes, where they only retained a regulatory function [97]. The case of *Phytophthora* is, thus, original and paves the path for future studies aiming at a better understanding of the mechanisms underlying the virulence of these pathogens.

## Figures and Tables

**Figure 1 microorganisms-10-00281-f001:**
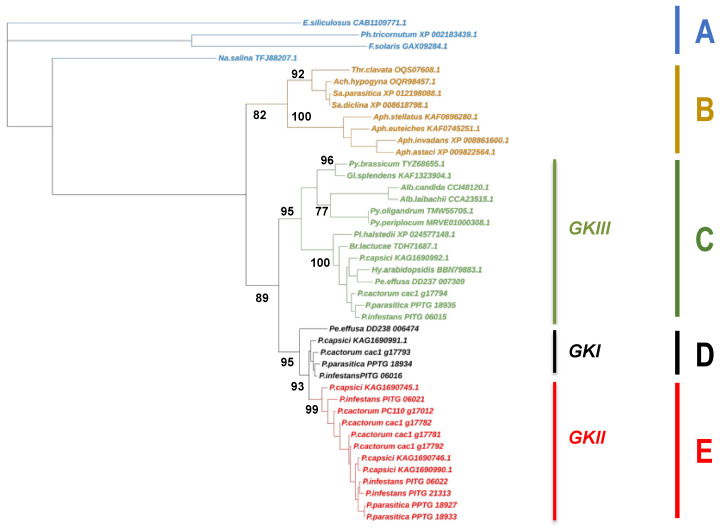
Phylogenetic tree depicting relationships among glucokinases from oomycetes. The tree was generated using MEGA and the maximum likelihood method. Sequences from brown algae and diatoms are indicated in blue. Oomycete sequences from Saprolegniales are presented in beige. Numbers at the nodes indicate bootstrap replicate values. Abbreviations are Ach.: *Achlya*; Al.: *Albugo*; Aph.: *Aphanomyces*; Br.: *Bremia*; E.: *Ectocarpus*; F.: *Fistulifera*; Gl.: *Globisporangium*; Hy.: *Hyaloperonospora*; Na.: *Nannochloropsis*; P.: *Phytophthora*; Pe.: *Peronospora*; Ph.: *Phaeodactylum*; Pl.: *Plasmopara*; Py.: *Pythium*; Sa.: *Saprolegnia*; and Thr.: *Thraustotheca*.

**Figure 2 microorganisms-10-00281-f002:**
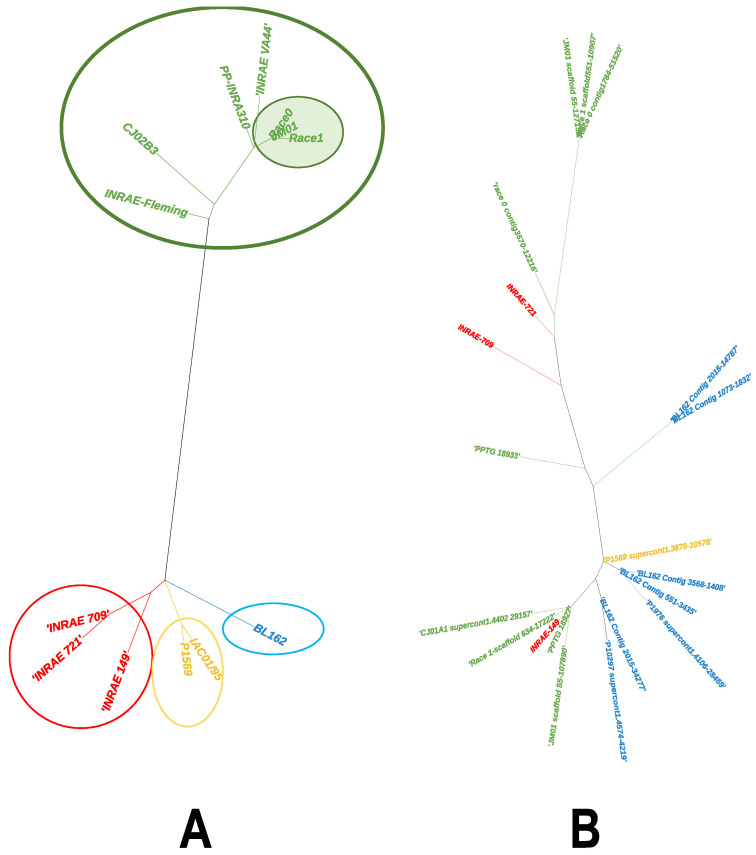
Intraspecific structuration of GKI and GKII sequences from *P. nicotianae*, as depicted by phylogenetic analyses. (**A**) Phylogenetic relationships among protein GKI sequences were established by maximum likelihood, using the Tamura-3-parameter substitution model, incorporating an among-site rate variation, approximated by a discrete gamma distribution (five rates). Branches were supported by 100 bootstrap replicates. Branch and label colors reflect the origin of sequences: green: tobacco; red: tomato; yellow: citrus; and blue: ornamentals. Tobacco isolates from China are identified by a green frame. (**B**) Phylogenetic relationships among GKII sequences were established by maximum likelihood, using the Kimura-2-parameter substitution model with a gamma (five rates) distribution. Branch and label colors are similar to those used for Figure 2A.

**Figure 3 microorganisms-10-00281-f003:**
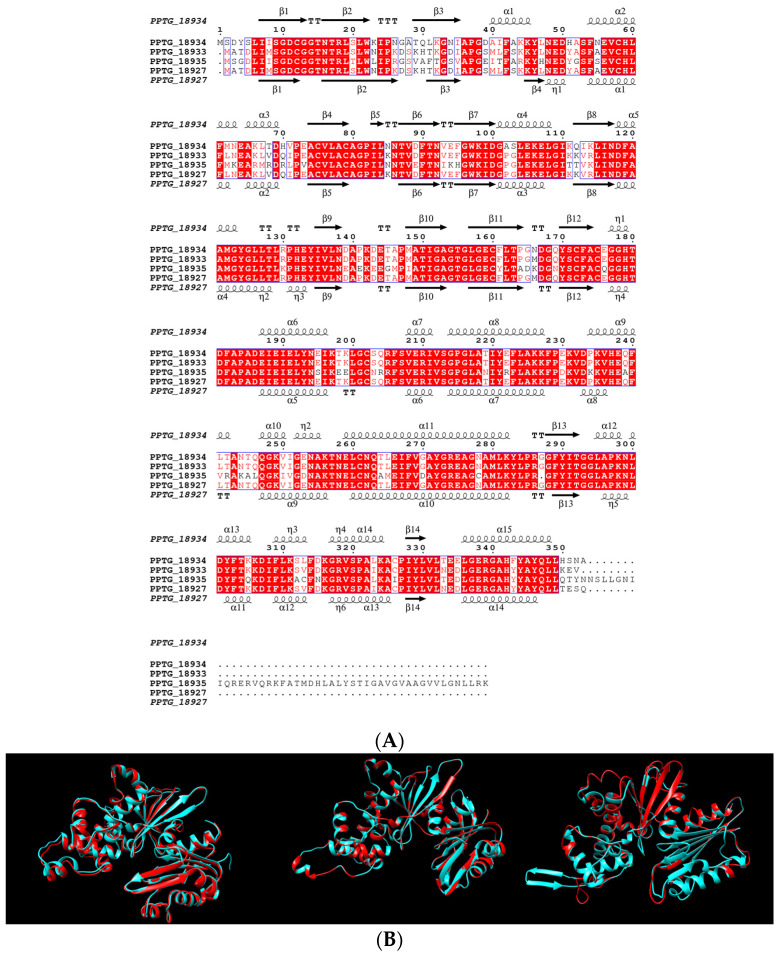
Secondary and 3D structures of *P. nicotianae* GKs. (**A**) Sequence- and structure-based alignment of *P. nicotianae* glucokinases. The alignment was generated by MUSCLE [41] and ESPript 3 [54], based on the predicted 2D structure of PPTG_18934 (GKI) and PPTG_18927 (GKII). Helices (α, η), sheets (β), and turns (TT) in the potential structure of GKI and GKII are labeled. (**B**) Superposition of I-TASSER predicted 3D model structures of *P. nicotianae* GKs (in blue) over their best analog templates (in red). Superposed are cartoons of GKI and *E. coli* glucokinase (PDB: 1sz2A, left), GKII and *Streptomyces* glucokinase (PDB: 3vpzA, middle), and GKIII and *N. fowleri* glucokinase (PDB: 6da0, right). Represented are cartoons of predicted 3D models of GKs, after structural refinement and energy minimization with ModRefiner. Pairwise structure alignments were performed using FATCAT, and images were rendered using ChimeraX.

**Figure 4 microorganisms-10-00281-f004:**
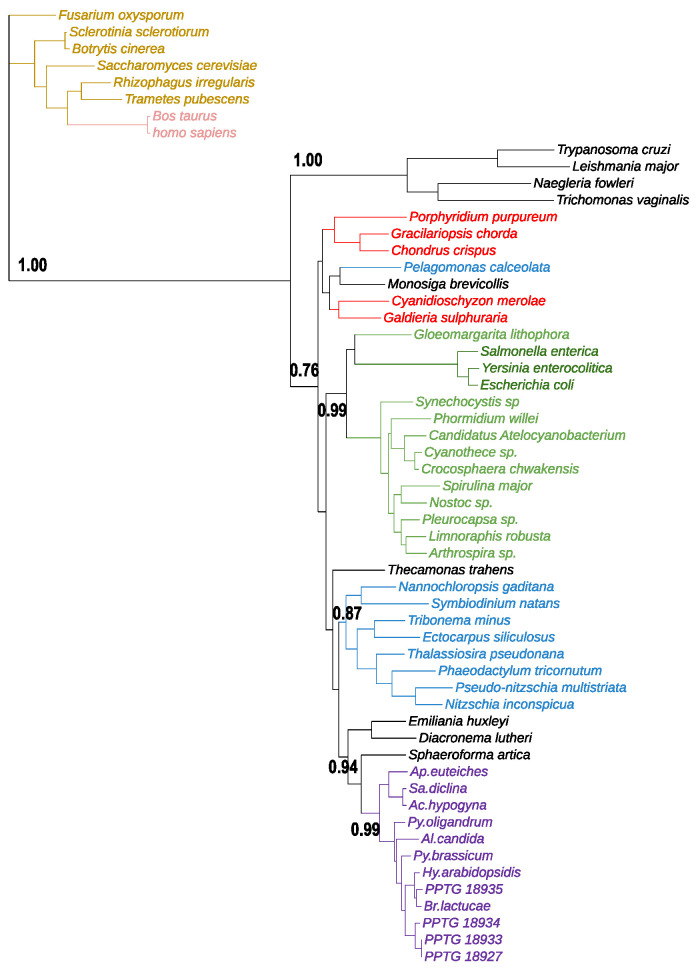
Unrooted maximum likelihood phylogeny of glucokinases. The tree was inferred with 57 protein sequences, retrieved from blastp searches, against the NR database at Genbank, using the four *P. nicotianae* GK proteins as queries. Sequences from oomycetes are presented in purple and other stramenopiles in blue. Bacteria (among which cyanobacteria) are represented in green, red algae in red, fungi in brown, and metazoan in pink. Less represented groups are described in the text. The tree was generated using MEGA and the maximum likelihood method. The numbers on branches indicate the results of the SH-like approximate likelihood ratio tests (aLRT). Branch support values larger than 0.7 were shown.

**Figure 5 microorganisms-10-00281-f005:**
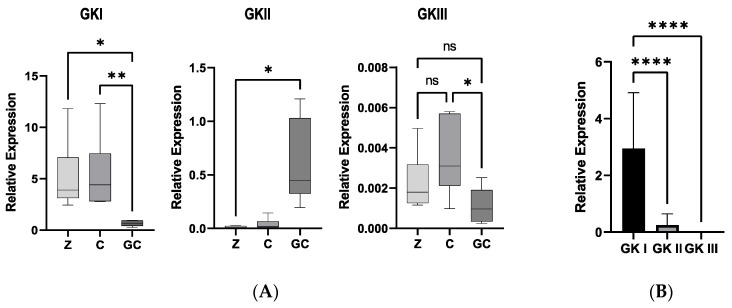
Expression patterns of GKs in pre-infection structures of *P. nicotianae*. (**A**) Values relative to each GK class were summed, normalized, and expressed, relative to the expression of the constitutive UBC gene [62]. (**B**) Expression levels of GKI, GKII, and GKIII in motile zoospores (Z), cysts (C), and germinating cysts (GC). Statistical analysis was performed in Graph Pad Prism v9.2 (GraphPad Software, San Diego, CA, USA), using one-way ANOVA and a Tukey’s multiple comparison tests. Statistical significance is denoted * *p* < 0.05, ** *p* < 0.01, **** *p* < 0.0001. ns: not significant.

**Figure 6 microorganisms-10-00281-f006:**
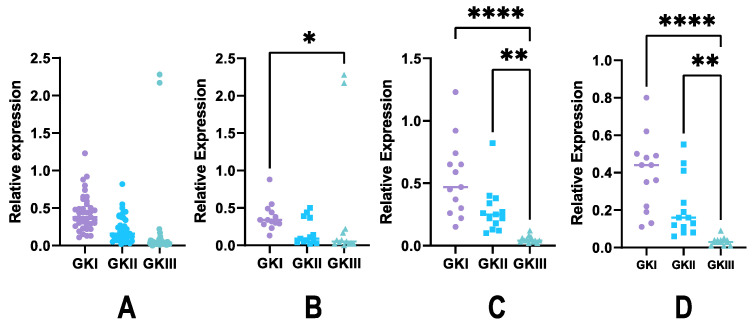
Expression of GK genes during plant infection. Roots of tomato and tobacco plantlets were inoculated with *P. nicotianae* zoospores, and expression of GK genes was evaluated relative to the expression of the constitutive UBC gene. Presented are expression values from samples after 2- (**A**), 4- (**B**), and 6-days (**C**) post inoculation. (**D**). Values from each kinetic experiment were pooled. Statistical analyses were conducted using a two-way ANOVA Friedman test. Statistical analysis was performed in Graph Pad Prism v9.2 (GraphPad Software, San Diego, CA, USA), using a two-way ANOVA Friedman test. Statistical significance is denoted * *p* < 0.05, ** *p* < 0.01, **** *p* < 0.0001.

**Figure 7 microorganisms-10-00281-f007:**
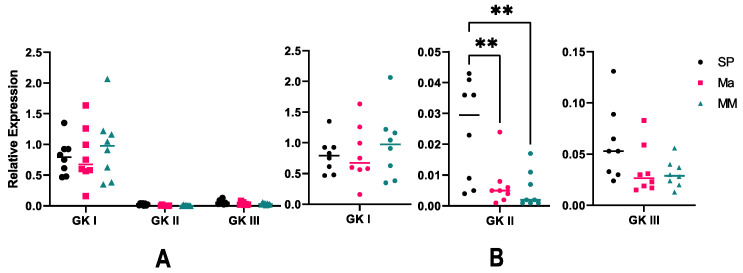
Relative expression of GK genes during leaf infection. Leaves from three tomato varieties were inoculated with zoospores of four *P. nicotianae* strains, and expression of GK genes was evaluated three-days post inoculation, relative to the expression of the constitutive UBC gene. (**A**) General view of expression values, as a function of tomato variety. (**B**) Representation of individual gene expression, for a better scale. SP: Saint Pierre; Ma: Marmande; MM: moneymaker. Statistical analysis was performed in Graph Pad Prism v9.2 (GraphPad Software, San Diego, CA, USA), using a two-way ANOVA Friedman test. Statistical significance is denoted ** *p* < 0.01.

**Figure 8 microorganisms-10-00281-f008:**
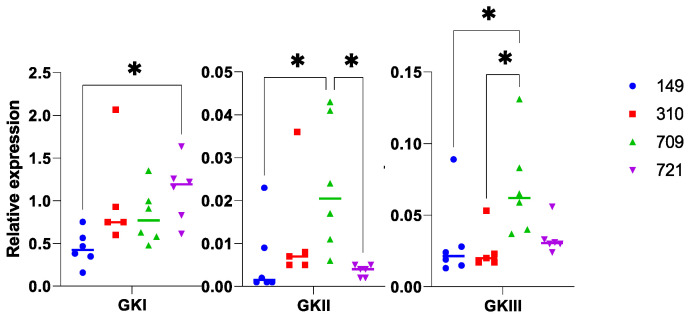
Relative expression of GK genes during leaf infection. Data correspond to the experiment, depicted in Figure 7, but are presented as a function of each *P. nicotianae* strain. Statistical analysis was performed in Graph Pad Prism v9.2 (GraphPad Software, San Diego, CA, USA), using a Kruskall-Wallis test. Statistical significance is denoted * *p* < 0.05.

**Figure 9 microorganisms-10-00281-f009:**
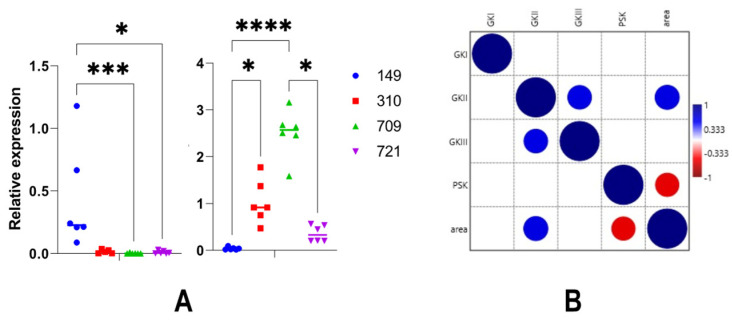
Expression of GKII is linked to *P. nicotianae* pathogenicity. (**A**) Comparison of the ability of the *P. nicotianae* strain to invade tomato leaves. Data refer to the same experiment than those analyzed in Figure 7 and Figure 8. Extent of infection was evaluated by the expression of a tomato gene, relative to the constitutive UBC gene (left), and measurement of invaded leaf surface, expressed in cm^2^ (right). Statistical analysis was performed in Graph Pad Prism v9.2 (GraphPad Software, San Diego, CA, USA), using a Kruskall-Wallis test. Statistical significance is denoted * *p* < 0.05, *** *p* < 0.001, **** *p* < 0.0001. (**B**) Correlation between the level of GK expression, the level of tomato gene expression, and extent of leaf invasion. Correlation values are represented by blue (positive) and red (negative) circles. The size of the circles reflects the numerical values. Analysis was conducted using a Kendal’s tau test using Past [64].

**Table 1 microorganisms-10-00281-t001:** Kinetic characteristics of the recombinant hexose kinases from *P. nicotianae*.

							Relative Activity with Phosphate Donors						
		*Km*	Glu	Glu 6-P	*Km*	*Km*							
	pH	Glucose	Inhibition	Inhibition	Fructose	ATP	ATP	UTP	CTP	GTP	ITP	PPi	ADP
*PPTG_18934*	8.0 ^1^	40.5 ± 5 µM	Y	N	NR	700 ± 50 µM	100	0.4	0.81	1,2	0	0	0
*PPTG_18927*	7.5 ^1^	32 ± 2.5 µM	N	N	NR	500 ± 50 µM	100	0.32	0.77	1	0.14	0	0.9
*PPTG_18935*	8.0 ^2^	125 ± 25 µM	N	N	NR	250 ± 25 µM	100	10.4	9.5	20.5	13.6	0	2.5
*PPTG_18886*	8.5 ^1^	NR	N	NR	57.5 ± 5 µM	1000 ± 50 µM	100	1.6	0.75	0.3	0.8	0	0

NR: not relevant. ^1^: phosphate buffer. ^2^: Tris buffer.

## Data Availability

This study is based on sequences available at GenBank. Unpublished data are available on request to the corresponding author.

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
