# Peer review of "Neofunctionalization of Glycolytic Enzymes: An Evolutionary Route to Plant Parasitism in the Oomycete *Phytophthora nicotianae"

_microorganisms, 2022, doi:10.3390/microorganisms10020281_

Round 1
Reviewer 1 Report
Microorganisms- “Neofunctionalization of glycolytic enzymes is an evolutionary route to parasitism in the oomycete Phytophthora parasitica”.
GENERAL COMMENT:
The work entitled “Neofunctionalization of glycolytic enzymes is an evolutionary route to parasitism in the oomycete Phytophthora parasitica” is a good work.
This study investigated the diversification of the glucokinases from P. parasitica and its possible consequences on the biology and pathogenicity of this pathogen.
The subject of the study is original, interesting and topical.
Central argument is supported by evidence and analysis.
The methodology described by the author is accurate.
Minor corrections throughout the text are needed in order to improve English language and punctuation.
This work only needs some minor changes, for this reason I require Minor Revision.
DETAILED COMMENT:
- Title
-The title is adequate but I suggest to emphasize it in the following way: “Neofunctionalization of glycolytic enzymes: an evolutionary route to parasitism in the oomycete Phytophthora parasitica”
- Abstract
-In the abstract the objective of the study is not clearly described. I suggest to improve it to make the purpose of the research clearer and immediately evident to the reader
Keywords
Keywords are adequate
- Introduction
The introduction section is not exhaustive and needs to be expanded. I suggest citing other previous works on the subject
- Materials and Methods
The section is well written and accurate.
- Results
This section is accurate and detailed
- Discussion
The discussion section is exhaustive and adequately discussed.
- Tables and figures
Tables and Figures are clear and understandable.
- References
The references are adequate
Author Response
General comment
We hope that the revised version addresses referee’s concerns. The whole text was revised for English language and punctuation. Here are some specific corrections.
Detailed comment
Title: we emphasized the title following referee’s suggestion.
-In the abstract the objective of the study is not clearly described. I suggest to improve it to make the purpose of the research clearer and immediately evident to the reader
We modified the abstract accordingly.
The introduction section is not exhaustive and needs to be expanded. I suggest citing other previous works on the subject
We modified the introduction, citing other works on the duplication of glycolytic genes in Phytophthora. We also introduced the notion of neofunctionalization of glycolytic enzymes, with possible impacts on promoted fitness and virulence of phytopathogens. We hope that these modifications will be satisfactory.
Reviewer 2 Report
Between having the wrong species name in the title and the naive understanding of oomycete evolution and systematics, I have no choice but to completely reject this manuscript. I am sure there is good data to present, but for a paper with an evolutionary context, a greater understanding of the literature is necessary before you re-submit.
The idea that Phytophthora evolved from an autotrophic ancestor is not a widely-held belief, and you cite a single source, and I personally disagree that the results of that source support this hypothesis. No genomes for the holoparasitic, early-diverging oomycetes exist, but a more nuanced story requires an acknowledgment of the marine, holoparasitic, early-diverging oomycetes. Furthermore, having the autotrophic diatoms and brown algae together in the stramenopiles does not necessarily mean a common autrotrophic stramenopile ancestor, since we know the photosynthetic apparatus was independently acquired through secondary endosymbiosis. An argument could be made that this endosymbiosis is still a form of parasitism.
Phytophthora parasitica is not the commonly-used name for this pathogen, and taxonomists agree that P. nicotianae has priority barring a taxonomic act. Change to P. nicotianae.
"oomycetes encompass four main orders: Saprolegniales, Albuginales, Pythiales and Peronosporales" Your reading of current oomycete higher-level taxonomy is very poor. Consult Beakes et al. 2014 (from "The Mycota"), but you've omitted at least seven orders from this list, and the Pythiales are no longer considered a valid order (see Thines and Choi 2016).
Your description of the Peronosporales omits Halophytophthora, Nothophytophthora and Calycofera as well as the entire Salisipiliaceae, which appear to be mainly saprotrophic.
The fact that you don't cite the literature demonstrating a wide variety of proteins (including potential targets of this paper) were horizontally transferred to oomycetes from true fungi is a major oversight. There are several papers but they start with Richards et al. 2011 doi/10.1073/pnas.1105100108, but the enzymes you're studying need to be classified as either oomycete-derived or ascomycete-derived.
Author Response
We agree that some taxonomical aspects were neglected in his study. The referee’s concerns induced a deeper investigation of the taxonomical relationships occurring among oomycetes and to a reappraisal of our phylogenetic analyses.
The idea that Phytophthora evolved from an autotrophic ancestor is not a widely-held belief, and you cite a single source, and I personally disagree that the results of that source support this hypothesis.
We agree with the referee that we cited a single source to support this statement, and that other hypotheses have been made. We modified this sentence to integrate these hypotheses.
Phytophthora parasitica is not the commonly-used name for this pathogen, and taxonomists agree that P. nicotianae has priority barring a taxonomic act. Change to P. nicotianae.
The denomination of P. parasitica or P. nicotianae is a long debate. We agree with the referee that P. nicotianae was coined first and may have priority. Yet, this name implicitly refers to tobacco, which is only one among the hundreds of hosts of this species and may be considered too restrictive. For this reason, the term P. parasitica is also commonly used by pathologists as it more accurately reflects the wide host range of this species. The two denominations are synonyms and are still used interchangeably, as illustrated by the number of reports from a search in Web of Science (559 for parasitica and 568 for nicotianae). We used both denominations in past articles (parasitica in ref 1, nicotianae in ref 8). We maintained the term parasitica in the present paper as it corresponds to the name of the reference genome mined in the study (P. parasitica PPINRA-310).
"oomycetes encompass four main orders: Saprolegniales, Albuginales, Pythiales and Peronosporales" Your reading of current oomycete higher-level taxonomy is very poor. Consult Beakes et al. 2014 (from "The Mycota"), but you've omitted at least seven orders from this list, and the Pythiales are no longer considered a valid order (see Thines and Choi 2016).
We did not pretend to do an exhaustive description of the current oomycete classification, and we only mentioned the main groups from which glucokinase sequences are available. Anyway, we modified the text according to the referee’s suggestion. We also abandoned the notion of Pythiales and adopted the current classification presented in more recent papers (Thines 2014, Thines and Choi 2016, Jung et al 2017).
Your description of the Peronosporales omits Halophytophthora, Nothophytophthora and Calycofera as well as the entire Salisipiliaceae, which appear to be mainly saprotrophic.
These organisms are now cited in the introduction.
The fact that you don't cite the literature demonstrating a wide variety of proteins (including potential targets of this paper) were horizontally transferred to oomycetes from true fungi is a major oversight. There are several papers but they start with Richards et al. 2011 doi/10.1073/pnas.1105100108, but the enzymes you're studying need to be classified as either oomycete-derived or ascomycete-derived.
We are grateful to the referee for this suggestion. In the present case, the most plausible hypothesis was that the donor of Phytophthora glucokinases would be of bacterial origin, as GK share several features with bacterial glucokinases (length, substrate specificity, sequence analogies…). We conducted additional searches in databases and phylogenetic analyses including a large set of sequences of various origins. We did not retrieve any hit with sequences from animals and fungi, despite the depth of our analyses, increasing the max target sequence parameter to 5000 in blast searches. It appears that glucokinases from Phytophthora branch with sequences from other Stramenopiles and may be classed as “oomycete-specific”. Results from this analysis have been organized into a new section (3.4).
Reviewer 3 Report
The authors have focused on Neofunctionalization of glycolytic enzymes as an evolutionary route to plant parasitism in the oomycete Phytophthora parasitica. In particular, the authors used a combination of phylogenetic, enzymatic, and transcriptional analyses to highlight that metabolic adaptation is a component of the processes underlying evolution of parasitism in Phytophthora, insinuating the possibility of neofunctionalization of metabolic enzymes. The authors are to be commended on their comprehensive analyses that hint on metabolic adaptation as a component of the processes underlying evolution of parasitism in Phytophthora.
The authors have successfully modelled the 3-D structures of P. parasitica GKs using the threading Iterative Threading ASSEmbly Refinement approach, however, further validation of the modelled structures needs to be done to determine the general stereochemical quality of the proteins.
Major
Much of the background information is on infection of Phytophthora spp. It is critical to highlight the concept of Neofunctionalization of glycolytic enzymes and how it promotes pathogen’s fitness and virulence in order to bring out the significance of the study in the introduction.
There is an association between subcellular localization of a protein and its function. The authors predicted the possible destination of P. parasitica glucokinases. However, the results for this analysis have not been mentioned.
Most of the stats done in the study revealed significant differences. It would be important for the authors to indicate (in the figure legend) the significance or confidence level (s) of their study.
Minor corrections
In lines 23, 65 and 671 the name Phytophthora has not been italicized. Same applies to P. parasitica in line 451. Supplementary Figure S5 can be made more useful by indicating the size of the proteins.
Line 152 ... consists 'of' remove in
Line 197 genome sequences (change genomes to genome)
In conclusion, while this manuscript contains much rich content that should be of interest to the readers of this journal, the reviewer suggests minor revisions of the manuscript to address the above raised comments. Hopefully, the authors will find my comments helpful in beginning to produce a more effective presentation of their interesting findings.
Author Response
Much of the background information is on infection of Phytophthora spp. It is critical to highlight the concept of Neofunctionalization of glycolytic enzymes and how it promotes pathogen’s fitness and virulence in order to bring out the significance of the study in the introduction.
We thank the referee for this suggestion. The introduction now deals with the notion of neofunctionalization as a means to improve Phytophthora fitness and possibly contribute to virulence. This topic has been also developed in the discussion.
There is an association between subcellular localization of a protein and its function. The authors predicted the possible destination of P. parasitica glucokinases. However, the results for this analysis have not been mentioned.
This remark is important, because it is directly connected to the question of the potential function of the different GKs. This point has been developed in the discussion.
Most of the stats done in the study revealed significant differences. It would be important for the authors to indicate (in the figure legend) the significance or confidence level (s) of their study.
The significance of stat is indicated now in each figure legend.
Minor corrections
In lines 23, 65 and 671 the name Phytophthora has not been italicized. Same applies to P. parasitica in line 451. Supplementary Figure S5 can be made more useful by indicating the size of the proteins.
Typographic corrections have been made. The Supplementary Figure S5 represents the in-gel revelation of enzymatic activity of FK and GK, so that electrophoresis was conducted under non-denaturing conditions. The precise size of proteins cannot be estimated accurately in this case. In addition, the GK activity detected here may correspond to the addition of the three or four predicted proteins which vary in length.
Line 152 ... consists 'of' remove in
Corrected.
Line 197 genome sequences (change genomes to genome)
Corrected.
In conclusion, while this manuscript contains much rich content that should be of interest to the readers of this journal, the reviewer suggests minor revisions of the manuscript to address the above raised comments. Hopefully, the authors will find my comments helpful in beginning to produce a more effective presentation of their interesting findings.
We found the comments of the referee very helpful and thank him/her very much for this review. We are aware that these comments helped us to produce an improved version of our manuscript.
Round 2
Reviewer 2 Report
I appreciate the changes that have been made based on my previous review of this manuscript. However, understanding that Phytophthora carries a large set of loci transferred from Ascomycota (Kingdom Fungi), you MUST include ascomycete (and other true-fungal) GKs in your phylogenetic tree. Because of this, I am completely dubious of your interpretation of your own results, which are otherwise very interesting. This manuscript needs a real revision, with some re-analysis, not just some changes to the introduction and discussion. When analyzing the sequences of the proteins, please indicated whether they were analyzed as amino acids, codons, or nucleotides.
Please refer to the organism as Phytophthora nicotianae. It is ridiculous to argue about this - one name is correct under taxonomic rules and the other is not, whether or not it happens to be a more apt name or still widely used. This journal has rules against publishing incorrect names, and this is one of the roles of peer review, to make sure they are followed. I am a worldwide Phytophthora expert and I have NEVER heard it referred to as P. parasitica in a modern discussion. Why not refer to P. cactorum as P. omnivora, since it is not always found on cactus? This is a ridiculous argument.
35 you have Stramenopiles capitalized. You are either referring to a Kingdom (Straminipila or Chromista) or a clade, but either way you need to clarify. either "the stramenopiles" or "Kingdom Straminipila" or "Kingdom Chromista"
35 admitted -> thought
47 Nothophytophthora is not an organism that has been recently revised, it was recently discovered, like Salisipilia. You are also leaving out Phytopythium of this discussion of the genera of the Peronosporales. Also downy mildews are definitely organisms whose "taxonomy was recently revised", so you really need to rework that paragraph
48 You just listed many genera, including several that are not known to be pathogenic (Salisipilia, Nothophytophthora, Halophytophthora). Then you refer to them collectively as pathogens. I appreciate your edits towards accuracy, but they have been done without care.
54 sudden oak death
55 The name of the species is not P. parasitica, it is P. nicotianae. As a Phytophthorologist, I am not going to accept this manuscript for publication with this arcane name.
67-67 allowed gaining insights -> reword
69-70 "deviations towards the canonical glycolysis" this doesn't make any sense as written
75 have two roles -> lead to several consequences
77 you need a citation for this statement, otherwise it is far too speculative
83 more citations!!
141 was the phylogeny performed on nucleotide or amino acid sequences? If a model testing was performed, what criterion was used, and what model was selected?
from the author response to my previous comments:
We did not retrieve any hit with sequences from animals and fungi, despite the depth of our analyses, increasing the max target sequence parameter to 5000 in blast searches. It appears that glucokinases from Phytophthora branch with sequences from other Stramenopiles and may be classed as “oomycete-specific”. Results from this analysis have been organized into a new section"
I do not personally believe that you conducted a thorough BLAST search. With what BLAST function were you searching? With the nucleotide or amino acid sequences of the protein? With the full length protein or sections?
A simple test search of "Saccharomyces" and "glucokinase" produces multiple hits, so I don't understand why you think you can't find fungal sequences for your phylogeny. If the fungal and metazoan glucokinases are so distinct from the ones you're studying in Phytophthora, this needs to be addressed in the introduction.
I cannot find the sequence accessions of the GKs you derived from the genomic data anywhere in the supplementary material, only references to the scaffolds and contigs. Therefore I cannot easily judge for myself whether this statement regarding the BLAST search is accurate.
Author Response
Point-by-point reply to referee
I appreciate the changes that have been made based on my previous review of this manuscript. However, understanding that Phytophthora carries a large set of loci transferred from Ascomycota (Kingdom Fungi), you MUST include ascomycete (and other true-fungal) GKs in your phylogenetic tree. Because of this, I am completely dubious of your interpretation of your own results, which are otherwise very interesting. This manuscript needs a real revision, with some re-analysis, not just some changes to the introduction and discussion. When analyzing the sequences of the proteins, please indicated whether they were analyzed as amino acids, codons, or nucleotides.
We performed some re-analysis as required, indicating that analyses of protein sequences were conducted using amino acids.
Please refer to the organism as Phytophthora nicotianae. It is ridiculous to argue about this - one name is correct under taxonomic rules and the other is not, whether or not it happens to be a more apt name or still widely used. This journal has rules against publishing incorrect names, and this is one of the roles of peer review, to make sure they are followed. I am a worldwide Phytophthora expert and I have NEVER heard it referred to as P. parasitica in a modern discussion. Why not refer to P. cactorum as P. omnivora, since it is not always found on cactus? This is a ridiculous argument.
We replaced parasitica by nicotianae throughout the text.
35 you have Stramenopiles capitalized. You are either referring to a Kingdom (Straminipila or Chromista) or a clade, but either way you need to clarify. either "the stramenopiles" or "Kingdom Straminipila" or "Kingdom Chromista"
We clarified this point in the text and replaced “Stramenopiles” by “the stramenopiles”.
35 admitted -> thought
We replaced “admitted” by “thought”.
47 Nothophytophthora is not an organism that has been recently revised, it was recently discovered, like Salisipilia. You are also leaving out Phytopythium of this discussion of the genera of the Peronosporales. Also downy mildews are definitely organisms whose "taxonomy was recently revised", so you really need to rework that paragraph
We reworked this paragraph according to referee’s suggestions.
48 You just listed many genera, including several that are not known to be pathogenic (Salisipilia, Nothophytophthora, Halophytophthora). Then you refer to them collectively as pathogens. I appreciate your edits towards accuracy, but they have been done without care.
We reworked this sentence accordingly.
54 sudden oak death
Corrected.
55 The name of the species is not P. parasitica, it is P. nicotianae. As a Phytophthorologist, I am not going to accept this manuscript for publication with this arcane name.
As indicated above, we replaced parasitica by nicotianae throughout the text.
67-67 allowed gaining insights -> reword
Corrected.
69-70 "deviations towards the canonical glycolysis" this doesn't make any sense as written
We reworked the sentence to make it clearer.
75 have two roles -> lead to several consequences
Corrected.
77 you need a citation for this statement, otherwise it is far too speculative
We included a citation supporting this hypothesis.
83 more citations!!
We added four citations supporting this statement.
141 was the phylogeny performed on nucleotide or amino acid sequences? If a model testing was performed, what criterion was used, and what model was selected?
The phylogenetic analyses were conducted on amino acid sequences. We apologize to have omitted to indicate the model that was selected for theses analyses. This is corrected now in the text. The best selected model was LG using a discrete gamma distribution (+G) with 5 rate categories.
from the author response to my previous comments:
We did not retrieve any hit with sequences from animals and fungi, despite the depth of our analyses, increasing the max target sequence parameter to 5000 in blast searches. It appears that glucokinases from Phytophthora branch with sequences from other Stramenopiles and may be classed as “oomycete-specific”. Results from this analysis have been organized into a new section"
I do not personally believe that you conducted a thorough BLAST search. With what BLAST function were you searching? With the nucleotide or amino acid sequences of the protein? With the full length protein or sections?
We estimate to have conducted a robust BLAST search. This was achieved using BLASTP and TBLASTN against NR with the full length sequences of PPTG_18934 (GKI) and PPTG_18935 (GKIII). As mentioned in our previous response, we increased the max target sequence parameter to 5000. We provide the hit lists (Suppl. File 1) of these searches showing that we did not retrieve any fungal sequence. Rather, the query sequences rapidly matched bacterial glucokinases, in frame with the other affinities to prokaryotic proteins already identified during this study (molecular weight, enzymatic properties…).
A simple test search of "Saccharomyces" and "glucokinase" produces multiple hits, so I don't understand why you think you can't find fungal sequences for your phylogeny.
- We agree with the referee that a simple text search of ‘Saccharomyces’ and ‘glucokinase’ produces multiple hits. We selected a set of these sequences for further analysis. We also conducted a specific blast analysis with Phytophthora GKs, restricting our search to “fungi”. “Yeast” and “fungal” sequences were integrated into a phylogenetic tree that is presented now in Figure 4 of the revised paper. We provide here an expanded tree with more sequences that show that “Fungal candidates” turned out to be from bacteria, as revealed by the tree topology (they are clustered with prokaryotic sequences (Suppl. File 2) and an additional Blast search against nr (Suppl. File 3). We hope that all these complementary data, that confirm our results and interpretations presented in the previous version of the paper, will convince the referee that we correctly conducted our searches.
If the fungal and metazoan glucokinases are so distinct from the ones you're studying in Phytophthora, this needs to be addressed in the introduction.
- This study reveals that fungal and metazoan ‘glucose kinases’ are distant from the ‘glucokinases’ from Phytophthora and other oomycetes. We indicated in the introduction that Phytophthora nicotianae possesses sequences of possible bacterial origin, instead of hexokinases (88-90), that are typically found in fungi and metazoans. We further state in the first section of the results (163-166) that :The predicted proteins possessed a molecular mass (38.65, 38.56, 38.59 and 43.64 kDa, respectively) that diverged from eukaryotic broad- substrate hexokinases [41], but that were similar to those calculated for prokaryotic, glucose-specific glucokinases [42], [43]. As it constitutes the first result of our analysis, it seemed logical to us to address it in the result section rather than in the introduction. Anyway, we modified the sentence in the last section of the introduction to stress out this point.
I cannot find the sequence accessions of the GKs you derived from the genomic data anywhere in the supplementary material, only references to the scaffolds and contigs. Therefore I cannot easily judge for myself whether this statement regarding the BLAST search is accurate.
All BLAST searches aiming at characterizing relationships between Phytophthora GKs with sequences from other organisms were conducted using the sequences publicly available at GenBank, identified with their PPTG_XXXXX nomenclature as queries. As well, phylogenetic trees presented throughout the paper contain these sequences. So, the referee can easily challenge our statements using the deposited sequences.
